# Critical Review on the Use of Extractives of Naturally Durable Woods as Natural Wood Protectants

**DOI:** 10.3390/insects15010069

**Published:** 2024-01-18

**Authors:** Grant T. Kirker, Babar Hassan, Mark E. Mankowski, Fred J. Eller

**Affiliations:** 1Durability and Wood Protection, USDA-FS Forest Products Laboratory, Madison, WI 53726, USA; 2Department of Agriculture and Fisheries, 50 Evans Road, Salisbury, QLD 4107, Australia; babar.hassan@daf.qld.gov.au; 3Durability and Wood Protection, USDA-FS Forest Products Laboratory, Starkville, MS 39759, USA; mark.e.mankowski@usda.gov; 4USDA, Agricultural Research Service, National Center for Agricultural Utilization Research, Functional Foods Research, 1815 N University, Peoria, IL 61604, USA; fred.eller@usda.gov

**Keywords:** extractives, natural products, wood protection, fungi, insects

## Abstract

**Simple Summary:**

Extractives, the non-structural component of woody biomass, are frequently targeted for their biocidal potential due to their evolutionary success in deterring pests in both standing trees and downed woody debris. Effective extractive utilization also offers an alternative product stream, where extractives are removed from the woody biomass that can further be used as feedstock for downstream processes (i.e., pulping, nanocellulose production, and biochar) once the extractives are removed. This review aims to provide details on prior studies using wood extractives as wood protectants, highlight the limitations to this approach, and discuss the research opportunities.

**Abstract:**

Naturally durable wood pre-dates preservative-treated wood and has been demonstrated to offer a suitable service life for certain applications where preservative-treated wood is not feasible. Heartwood extractives have been demonstrated to impart bio-deteriorative resistance to naturally durable wood species. These extractives are typically found in the heartwood of living trees and are produced either by the death of parenchyma cells or as the result of external stimuli. The mechanisms of natural durability are not well understood, as heartwood extractives can be extremely variable in their distribution, composition, and efficacy in both living and harvested trees. The underlying complexity of heartwood extractives has hindered their standardization in residential building codes for use as wood preservatives. The use of naturally durable lumber is not always feasible, as woods with exceptionally durable heartwood do not typically yield lumber with acceptable machining properties. A potential approach to overcome the inherent difficulty in establishing guidelines for the appropriate use of naturally durable wood is to focus solely on the extractives as a source of bioactive protectants based on the strategies used on living and dead wood to repel the agents of biodeterioration. This critical review summarizes the relevant literature on naturally durable woods, their extractives, and their potential use as bio-inspired wood protectants. An additional discussion will be aimed at underscoring the past difficulties in adopting this approach and how to overcome the future hurdles.

## 1. Introduction

Wood is a sustainable building material with a low cost, high strength-to-mass ratios, and favorable carbon benefits compared to those of the competing materials [1]. As a biomaterial, wood subjected to frequent wetting or adverse conditions (i.e., ground contact and marine exposure) is subject to biodegradation through the actions of decay fungi or wood-attacking arthropods [2]. For the aforementioned scenarios, pressure treatment with an approved chemical wood preservative is typically the best choice, especially for the critical wood members contributing to structural integrity. More recently, chemically and thermally modified woods are gaining popularity worldwide, especially in areas with heightened regulations on chemically impregnated wood and/or limited access to durable wood species [3]. However, there are certain situations where pressure-treated or modified wood is not recommended. Naturally durable woods can be used in these applications, such as exterior siding, livestock fencing, decking, or for other less structurally critical interior and exterior uses.

The history of using durable timber parallels the history of wood utilization. Humans learned early to select wood for construction and wooden tools to last for a long time. Mine timbers, ship masts, fence posts, and cribbing are examples of wooden commodities with a documented history of exceptional service and suitability using naturally durable wood species. Only since the middle of the last century has the importance of naturally durable timber species decreased inversely to the increasing development of industrial timber impregnation [4]. However, biocidal wood preservation has recently been in the spotlight due to environmental concerns that have led consumers to reconsider more environmentally friendly alternatives, including the use of naturally durable wood. Several factors, such as specific gravity, density, water exclusion efficiency, and, most importantly, extractives, make wood naturally durable [5,6]. Extractives are usually most highly concentrated in heartwood, and can contain many active chemical compounds (terpenes, stilbenes, resin acids, tropolones, antioxidants, etc.), which impart resistance to wood-destroying organisms either alone or in synergy with the other wood chemical extractives collocated within the heartwood of a tree. The mechanisms of naturally durable wood have been reviewed several times [7,8,9] and are outside the scope of this review. The goal of this review is instead to focus on existing work that has attempted to use heartwood extractives to improve the resistance of less-durable wood species through direct impregnation (transferred durability) against termites and decay fungi; discuss the significant findings, limitations, and the outcomes of the work; and propose a future framework for advancing the field of extractive utilization as bioinspired wood protectants.

### 1.1. Commercial Examples of Naturally Durable Wood

Many durable timber species are widely recognized due to their well-known durability and accompanying structural quality. Table 1 gives popular examples of naturally durable hardwoods and softwoods that are commercially available worldwide. However, the natural forest resources in many parts of the world are depleting at an alarming rate, threatening the extent of availability of many of these naturally durable timbers, and the cost of obtaining useful dimensions of certain species limits their utility. In addition, some durable wood species, such as *Pericopsis elata*, *Vouacapoua americana*, *Pterocarpus angolensis*, etc., are almost extinct and/or protected by international entities, such as the Convention on International Trade of Endangered Species (CITES) and the International Union for Conservation of Nature (IUCN). Due to the incongruency of rating scales between the various durability classifications (EN350–2 [10], Australia, IBCC, etc.), durability is presented as a range from highly durable to non-durable for all the species presented in Table 1 and Table 2.

### 1.2. Standardization of Naturally Durable Wood

There are currently no listings or standardizations in North America regarding the end uses of natural durable wood species as there are for preservative-treated wood. However, the International Building Code (IBC) does specify the use of heartwood timbers from redwood, cedars, black walnut and black locust for exterior, above-ground applications, as well as for most ground-contact applications. However, the IBC does not differentiate between the old growth and second growth of these species; the former of which have been shown to contain greater proportions of heartwood extractives, as well as material density [7]. Additionally, the IBC does not specify using heartwood processed from imported wood species.

Internationally, the Australian AS 5604 standard provides natural durability classifications for untreated timber for decay in and above ground, termite and marine borer resistance, and lyctus susceptibility. While the Australian National Construction Code–Building Code of Australia (BCA) currently does not have specific durability performance requirements, it does have implicit requirements for using naturally durable or treated wood species. The Australian Building Codes Board has published a wood durability guideline document that guides the manufacturers, users, appraisers and others on the implicit durability performance requirements of treated and naturally durable wood species [14]. Similarly, the European standards EN 350–1 and EN 350–2 are used as reference documents on wood species’ natural resistance against decay fungi, wood-boring beetles, and termites in Europe [10]. Part 1 lists the methods for determining the natural resistance of untreated solid wood to attacks by wood-destroying organisms and the principles of wood species classification based on the results of test methods. Part 2 lists the natural durability and treatability of important wood species in Europe. At the same time, the EN 460 guidelines for the durability requirements for wood to be used in different hazard classes are defined in EN 335 [15].

### 1.3. Factors Limiting the Effective Use of Naturally Durable Wood

The extractives deposited during the formation of heartwood are the principal source of resistance to wood-destroying organisms. The variability in extractive content is and has always been the limiting factor in establishing effective use guidelines for naturally durable wood. The decades of research have aimed to elucidate the patterns in genetic and physiological bases for the natural durability of wood [7,16,17,18]. The distribution of decay resistance within a tree stem has been correlated with the distribution and nature of toxic extractives [9]. Moreover, the natural durability of wood strongly varies according to the geographic regions, environmental exposure conditions of growing trees, individual trees, silvicultural practices, and age of the trees [19]. The wood used in tropical climates generally deteriorates much faster than the same wood species used in temperate regions [20]. Lastly, wood durability can be diminished in the second-growth and plantation material of the same tree species [21].

The protection provided by wood extractives is not permanent; the extractives involved can be denatured or lost from the wood over time [22], which can also occur, albeit over much longer time spans, in preservative-treated wood. In some cases, chemical extractions have indicated that some extractives may be physically locked in the wood, and thus, may resist this loss through leaching. However, the other compounds in wood extractives are water-soluble and can readily leach out over time, making the wood less durable [23,24]. Similarly, lignin and extractive degradation due to exposure to UV light may reduce the durability of wood [25]. The losses through evaporation are quite minor, as indicated by the small reduction in decay resistance produced by dry heat; however, this facet of change in natural durability warrants further study [9,26,27]. The microbial degradation of extractives is another mechanism of change in durability. The decay-inhibiting extractives are depleted due to chemical changes caused by non-decay fungi that invade the heartwood when they are exposed to the great variety of organisms present in soil [9,28].

Additionally, some supplies of naturally durable woods, particularly those of tropical origin, have significantly declined over the years, which could be another limiting factor in the use of durable wood [29]. An important consideration in the use of tropical woods is that of sustainability and environmental stewardship. Valuable heartwood extractives are contained in much of the old growth forest ecosystems around the globe, but further stresses to those ecosystems could contribute further to the effects of changing climate. Many of the wood species that exhibit excellent heartwood durability and history of use are components of rainforest ecosystems, and their use should be limited to avoid further deforestation of this important carbon sink. The desire to limit the use of old-growth tropical timbers creates new opportunities in wood utilization that could be more focused on using the chemical components within forest residues (i.e., limbs, bark, roots, and mill waste), such as heartwood extractives, where these components are synthesized as to be available on a commercial scale.

### 1.4. Extractives as Wood Protectants

Given that heartwood extractives have been shown to impart a considerable proportion of the inherent durability to naturally durable wood species, the extractives vary widely within and among species. Additionally, durable species do not always present growth habits conducive to marketable timber; a logical approach is to remove the extractable components in order to transfer them to a non-durable wood with better workability. This is normally accomplished by using a suitable solvent (based on the solvent’s polarity, volatility, miscibility, etc.) to remove the heartwood extractives from the structural matrix of the wood that is leached out into a liquid phase [30]. This concept predates the use of industrial wood preservatives, as pine tars were used as early as the 17th century in Scandinavia to preserve the wood on maritime vessels, which serves as one of the earliest attempts to preserve wood. Pine tars from Scots pine combined with linseed oils were used as brush-on applications, as well as to produce oakum that was used in the waterproofing and sealing of ship hulls [31]. This practice was carried over into the colonial American continents and eventually gave rise to the naval stores industry, which thrived in the United States up until the early 1900s [32] The more recent studies have shown that these pine tars contain fatty and resin acids, sterols, stilbenes, steryl esters, and lignans [33,34].

The presence of heartwood extractives is considered a nuisance when producing pulp for paper products, as the extractives impart undesirable color and confound the papermaking process [35]. Several modern procedures are used for the extraction, isolation, and identification of extractives that have been well explained [36,37]. Briefly, the air-dried heartwood is ground to a powder that is extracted with Soxhlet apparatus or a column using different solvents of increasing polarity or with a single solvent. The choice of extraction system or solvent depends on the chemical nature of the extractives being removed [38]. Once the extract has been obtained, it may be used in a crude form by dissolving or diluting in the respective solvent, or it can be further divided into fractions containing compounds of a similar chemical nature. If possible, the isolated and identified extractives are used to treat non-durable wood. Laboratory and field tests on the efficacy of commercial wood preservatives are used to test an extractives’ wood protection potential, and these candidate preservative systems are compared to the existing systems with established performance criteria. Like in any bioassay, efficacy depends on the compounds/extractives being tested, their concentration, the organism being studied, and the conditions under which the test is performed. Vacuum-pressure impregnation is the most common method employed for the uniform penetration of extractives into non-durable wood species. This process is the standard method to ensure adequate amounts of bioactive compounds are incorporated into the wood, and several variations exist based on the process, vacuum/pressure schedule, and the severity and duration of the pressure or vacuum [39,40]. Weight loss is the standard measure used to quantify the damage rate once the extractives have been impregnated into the wood and exposed to test fungi and/or termites. However, damage rating schemes specified by different standardized tests are used to assess the wood exposed in the field [41].

The use of heartwood extractives as wood protectants has been the focus of several decades of research and is summarized in Table 2. This list of studies includes only those that pertain to the use of extractable compounds from wood and the subsequent treatment of wood to improve its service life. For more general reviews on composition and biological activity of heartwood extractives, there are several excellent reviews on this subject [7,42,43]. The type of test is indicated as either laboratory (L) or field (F), and the expected durability classifications are based on these bibliographical references [11,44].

**Table 2 insects-15-00069-t002:** Example of studies on use of extractable compounds from wood and subsequent treatment of non-durable wood to improve its resistance against termites and decay fungi.

Durable Wood	Durability Class * [11,12,13]	Biodeterioration Organism	Non-Durable Wood	Treatment Method	Type of Test **	Origin of Durable Wood [11]	Reference
*Bagassa guianensis*, *Manilkara huberi*, *Sextonia rubra*, *Vouacapoua Americana*, *Andira surinamensis*, *Handroanthus serratifolius*, *Qualea rosea*	1, 2, 1–2, 1–2, 2–3, 2, 3	*Gloeophyllum trabeum*, *Trametes versicolor*, *Reticulitermes flavipes*, *Nasutitermes macrocephallus*	*Pinus sylvestris*, *Virola surinamensis*	Vacuum-pressure impregnation, brushing	L	Tropical America	[45,46]
*Afzelia Africana*, *Callitris glaucophylla*	1–2	*Coridopsis polyzona*, *Lenzites trabea*, *Trametes cingulate*, *Lopharia crassa*, *Polyporus verecundus*	*Antiaris toxicaria*, *Pinus radiata*	Vacuum-pressure impregnation	L	Africa, Australia	[47,48]
*Tectona grandis*	1	*Heterotermes indicola*, *Postia placenta*, *Neolentinus lepideus*, *Gloeophyllum trabeum*, *Coptotermes curvignathus*, *Nasutitermes corniger*, *Cryptotermes brevis*, *Trametes versicolor*, *Nasutitermes* sp., *Heterotermes* sp., *Polyborus sanguineus*, *Schizophillum commune*, *Pannus crinitus*, *Poria placenta*, *Gmelina arborea*, *Triplochiton scleroxylon*	*Pinus* sp., *Populus* sp., *Pinus patula*, *Eucalyptus globulus*, *Pinus sylvestris*, *Gmelina arborea*, *Triplochiton scleroxylon*	Vacuum-pressure impregnation, Dipping	L, F	Asia	[49]
*Dalbergia sissoo*	1	*Heterotermes indicola*, *Reticulitermes flavipes*, *Postia placenta*, *Trametes versicolor*	*Pinus* sp., *Populus* sp.	Vacuum-pressure impregnation	L, F	Asia	[49]
*Cedrus deodara*	3	*Heterotermes indicola*, *Reticulitermes flavipes*, *Postia placenta*, *Trametes versicolor*	*Pinus* sp., *Populus* sp.	Vacuum-pressure impregnation	L, F	Asia	[49,50]
*Pinus roxburghii*	4	*Heterotermes indicola*, *Reticulitermes flavipes*, *Postia placenta*, *Trametes versicolor*	*Pinus* sp., *Populus* sp.	Vacuum-pressure impregnation	L, F	Asia	[49]
*Morus alba*	4	*Postia placenta*, *Trametes versicolor*, *Heterotermes indicola*, *Reticulitermes flavipes*	*Pinus* sp., *Populus* sp., *Fagus orientalis*, *Acer insgin*, *Alnus subcordata*, *Tilia* sp.	Vacuum-pressure impregnation	L	Asia	[51,52]
*Morus nigra*	NA	*Heterotermes indicola*	*Populus* sp.	Pressure impregnation	L	Asia	[51]
*Ziziphus mauritiana*	NA	Termites	*Populus deltoides*	Dipping	F	Asia	[53]
*Milicia excelsa*, *Albizia coriaria*, *Markhamia lutea*	1–2	*Macrotermes bellicosus*, *Coridopsis polyzona*, *Lenzites trabea*, *Trametes cingulate*	*Pinus caribaea*, *Antiaris toxicaria*	Dipping	F, L	Africa	[47,54]
*Erythrophleum suaveolens*, *Chlorophora excelsa*	1–2, 1	*Coridopsis polyzona*, *Lenzites trabea*, *Trametes cingulate*	*Antiaris toxicaria*	Dipping	L	Africa	[47]
*Acacia mollissima*, *Shinopsis lorentzii*	4	*Reticulitermes grassei*	*Pinus sylvestris*, *Fagus orientalis*, *Populus tremula*	Vacuum-pressure impregnation	L	Asia	[55]
*Acacia mearnsii*	NA	*Pycnoporus sanguineus*	*Acacia mearnsii*	Vacuum-pressure impregnation	L	Australia, Europe, America	[56]
*Juniperus virginiana*	1–2	*Reticulitermes flavipes*, *Gloeophyllum trabeum*, *Postia placenta*, *Irpex lacteus*, *Trametes versicolor*	*Pinus* sp., *Picea* sp., yellow poplar	Vacuum-pressure impregnation	L	Temperate America	[57,58]
*Cupressus nootkanansis*, *Prosopis glandulosa*, *Robinia pseudoacacia*	-, -, 1–2	*Reticulitermes flavipes*, *Postia placenta*, *Gloeophyllum trabeum*, *Schizophyllum commune*, *Fibroporia vaillantii*	*Pinus taeda*, Aspen, *Fagus sylvatica*	Vacuum-pressure impregnation, vacuum only	L	Temperate America	[59,60,61]
*Chamaecyparis lawsoniana*, *Catalpa bignonioides*	1–2	*Reticulitermes flavipes*	*Liquidambar styraciflua*, *Pinus* spp.	Vacuum-pressure impregnation	L	Temperate America	[62,63]
*Callitris columellaris*	3	*Coptotermes acinaciformis*, *Nasutitermes exitiosus*, *Mastotermes darwiniensis*	*Eucalyptus regnans*	Dipping	L	Asia, Australia	[64]
*Pterocarpus soyauxii*	1	*Poria placenta*, *Trametes versicolor*, *Gloeplyllum trabeum*, *Coniophora puteana*, *Coriolus versicolor*	Aspen wood, *Pinus sylvestris*, *Fagus sylvatica*	Vacuum dipping	L	Africa	[61,65,66]
*Chamaecyparis nootkatensis*, *Juniperus occidentalis*	2–3, 1	*Gloeophyllum trabeum*, *Postia placenta*, *Irpex lacteus*, *Trametes versicolor*	*Pinus* sp., *Picea* sp., yellow poplar	Vacuum-pressure impregnation	L	Temperate America	[67]
*Catalpa speciosa*, *Paulownia tomentosa*	NA	*Gloeophyllum trabeum*, *Irpex lacteus*, *Postia placenta*, *Trametes versicolor*	Southern pine	Vacuum-pressure impregnation	L	-	
*Juniperus ashei*	NA	*Gloeophyllum trabeum*, *Postia placenta*, *Irpex lacteus*, *Trametes versicolor*	*Pinus* sp., Picea sp., yellow poplar	Vacuum-pressure impregnation	L	Temperate America	[67]
*Tabebuia* sp.	1–2	*Trametes versicolor*, *Fomitopsis palustris*	*Cryptomeria japonica*, *Fagus* sp.	Pressure impregnation	L	Tropical America	[68,69]
*Breonadia salicina*, *Spirostachys Africana*, *Syncarpia glomulifera*, *Paeroxylon obliquum*	1–2	Termites and decay fungi	*Pinus patula*	Vacuum-pressure impregnation	F	Africa	[70]
*Nauclea diderrichii*	1	*Coniophora puteana*, *Coriolus versicolor*, *Pleurotus ostreatus*, *Gloeophyllum sepiarium*	*Pinus sylvestris*, *Triplochiton scleroxylon*	Vacuum dipping	L	Africa	[66]
*Aniba rosaeodora*	1	*Reticulitermes santonensis*, *Reticulitermes flavipes*	*Pinus* sp.	-	L	Tropical America	[71,72]
*Hopea parviflora*	1	Termites, *Polyporus meliae*, *Tyromyces palustris*, *Pycnoporus sanguineus*, *Coriolus hirsutus*	*Gyrocarpus jacquinii*, *Mangifera indica*	Dipping	L, F	Asia	[73]
*Zelkova carpinifolia*, *Quercus castanifolia*, *Fraxinus excelsior*	2–4	*Trametes versicolor*	*Fagus orientalis*, *Acer insgin*, *Alnus subcordata*, *Tilia* sp.	Vacuum-pressure impregnation	L	Asia	[52,74]
*Juglans regia*	4	*Coriolus versicolor*, *Trametes versicolor*	*Poplus* spp., *Fagus orientalis*	Vacuum dipping, dipping	L	Asia	[75,76]
*Eusideroxylon zwageri*, *Potoxylon melagangai*	1, -	*Pycnoporus cocinneus*, *Schizophyllum commune*	*Hevea brasiliensis*	Dipping	L	Asia	[77]
*Eucalyptus sideroxylon*	1	*Polyporus versicolor*, *Poria monticola*	*Populus deltoides*	Vacuum dipping	L	Australia	[78]
*Maclura pomifera*, *Sequoia sempervirens, Intsia bijuga*	2, 1–3	*Gloeophyllum trabeum*	*Populus tremuloides*	-		Temperate America, Asia, Australia	[79]
*Pinus sylvestris*	4	*Postia placenta*, *Gloeophyllum trabeum*, *Schizophyllum commune*, *Fibroporia vaillantii*, *Coniophora puteana*	*Fagus sylvatica*, *Pinus sylvestris*	Vacuum-pressure impregnation	L	Europe	[80]
*Neobalanocarpus heimii*, *Cotylelobium lanceolatum*, *Madhuca utilis*, *Shorea curtisii*	1, 1–2, 1, 4	*Coptotermes curvignathus*, *Coptotermes gestroi*, *Trametes versicolor*, *Lentinus sajor-caju*, *Coniophera puteana*	*Hevea brasiliensis*	Vacuum dipping	L	Asia	[81]
*Cinnamomum* sp., *Canarium littorale*, *Cynometra malaccensis*, *Eugenia griffithii*, *Scorodocarpus borneensis*	2–4, 4, 4, 2–3, 3	*Coptotermes gestroi*	*Hevea brasiliensis*	Vacuum-pressure impregnation	L	Asia	[82]
*Mansonia altisima*	1	*Pleurotus ostreatus*, *Gloeophyllum sepiarium*, *Gloeophyllum* sp.	*Triplochiton scleroxylon*	Dipping	L	Africa	[83]

* Durability classification scale: 1 = very resistant; 2 = resistant; 3 = moderately resistant; 4 = non-resistant. ** L = Laboratory test; F = Field tests.

## 2. Discussion

### 2.1. Factors Limiting the Use of Wood Extractives as Wood Protectants

Wood extractives are mixtures of toxic and nontoxic components that interact with one another in the wood to provide resistance against biodeterioration [7]. Sometimes, impregnated non-durable wood is not as resistant as the original piece of durable wood. For example, when stilbenes from pine, Osage orange, and eucalypt wood were impregnated into non-durable wood, their efficacy against decay fungi was reduced to ~1/100 of the original block of heartwood [84]. An important distinction here is the difference in the spatial location of extractives that are naturally occurring versus those that are artificially placed. In a living tree, the extractives are produced as a wound or invasion response and are highly localized based on individual need of the tree [7], and in an artificially impregnated material, the extractives are likely more widespread, homogenized, and diluted compared to their native state. In contrast, the toxicities of ellagitannins and tropolones in the unextracted heartwood of *Eucalyptus sideroxylon* A. Cunn. ex Woolls and impregnated into non-durable wood at their original concentrations were very similar [78]. Therefore, impregnating non-durable wood with heartwood extractives sometimes does not precisely duplicate the situation in the original tree. No data exist showing that the location and form of extractives in the original heartwood and artificially impregnated woods are the same.

The binding between the woody components and extractives may differ in naturally and artificially impregnated wood [38]. Moreover, the complete removal of extractives from wood is sometimes impossible, and it depends on the solvent used and the other extraction conditions. Some extractives can also react with lignin and form an extractive–lignin complex. They are not extractable and are even more critical in ensuring wood durability. This explains the phenomenon of wood durability after artificial weathering or extensive leaching [37]. Wood resistance against biodeterioration is a multifunctional phenomenon. It may be impossible to confer on one substance possessing sole responsibility for the entire heartwood’s resistance; the other factors, including wood density or lignin contents, may also be involved and act synergistically [9]. Therefore, testing the extractives singly or as a mixture may yield misleading results.

### 2.2. Regulatory Hurdles

Contrary to the public perception, chemically treated wood is a highly regulated and controlled commodity. In the United States, the wood preservatives are regulated through the FIFRA (Federal Insecticide and Fungicide Registration Act). Under the FIFRA, all the industrial and residential wood preservatives are regulated from formulation to application and are subject to review every 15 years. At re-evaluation, these compounds are reviewed to determine if they pose significant risks to the environment, public, or workers that outweigh their utility. The most recent regulatory changes in the US would be the removal of pentachlorophenolin in 2021 as an industrial wood preservative, and additional safety mitigation methods were applied to chromated arsenicals and creosote. However, they are still allowed for use as wood preservatives for at least 15 more years [85]. A decreasing arsenal of effective wood preservatives emphasizes the need for research into new protective strategies to prolong the useful service life of wood and new chemical classes of protectants based on extractives that could serve as scaffolds for bio-inspired pesticides. Internationally, the Biocidal Product Regulation 528/2012, formerly known as Biocidal Directive 98/8/EC, authorizes the trade and use of wood preservatives in Europe [86]. Biocides are approved for 5 or 10 years, and authorization requires an acceptance procedure, where the risks to the environment and human health are assessed. The compounds containing mutagens; endocrine-disrupting, persistent, and bio-accumulative elements; carcinogens, etc., do not receive authorization. REACH (Registration, Evaluation, Authorization and Restriction of Chemicals) is responsible for establishing the procedures for collecting and assessing information on the properties and hazards of biocides to be registered. Then, the European Chemical Agency (ECHA) coordinates the registration of biocidal products and provides EU-wide authorization. National authorization can also be applied [76]. Similarly, in Australia, the Australian Pesticides and Veterinary Chemical Authority (APVMA) is the national biocidal regulator responsible for approving, regulating, and labeling the biocides used to treat timber. The Environmental Protection Authority (EPA) regulates the wood preservation industry and sets the standards for the responsible use of timber treatments [87]. There are extensive data requirements for chemical antimicrobials that include environmental fate, human health, and ecological data to ensure that the new and existing preservatives do not pose a significant risk to the environment and human health. The Environmental Protection Agency (EPA) has jurisdiction over antimicrobial formulations in the US, but wood extractives would be more suitably regulated under “biopesticides”. Biopesticides are defined as pesticides derived from natural materials, such as animals, plants, bacteria, and certain minerals. These are also regulated under the FIFRA, but are subject to different registration guidelines than the conventional chemicals. These requirements are listed in Data Requirements for Registration (40 CFR Part 158). Certain organisms are exempt from FIFRA guidance, including pheromones, insect predators, mites, nematodes, entomopathogens and botanicals.

The minimum-risk pesticides, as defined by the EPA, must satisfy six guidance criteria:Condition 1: The product’s active ingredients must only be those that are listed in 40 CFR 152.25(f)(1).Condition 2: The product’s inert ingredients may only be those that have been classified by EPA as:
⚬Listed in 40 CFR 152.25(f)(2);⚬Commonly consumed food commodities, animal feed items, and edible fats and oils as described in 40 CFR 180.950(a), (b), and (c);⚬Certain chemical substances listed under 40 CFR 180.950(e).Condition 3: All the ingredients (both active and inert) must be listed on the label. The active ingredient(s) must be listed by label display name and percentage by weight. Each inert ingredient must be listed by its label display name.Condition 4: The product must not bear claims either to control or mitigate the organisms that pose a threat to human health, or insects or rodents carrying specific diseases.Condition 5: The name of the producer or the company for whom the product was produced, and the company’s contact information must be displayed prominently on the product label.Condition 6: The label cannot include any false or misleading statements [88].

Outside of the United States, there are several important regulatory bodies that govern the approval and use of natural products. For example, in Australia, the APVMA regulates biological chemical products where the active constituent comprises or is derived from a living organism (plant, animal, micro-organism, etc.), with or without modification. This also includes unpurified or partially purified extracts derived from plants, including oils or other extracts. Similarly, in Europe, biocides containing plants, parts of plants, or plant products are approved by the European Commission [89]. Like other biocides, the registration of extractives-based wood preservatives requires extensive testing before approval.

## 3. Concluding Remarks

Extractive utilization presents an excellent opportunity to harvest the non-structural components of wood to be used as value-added forest products. In this review, only the utility of extractives as bioprotectants is discussed, but the market potential could be far more significant once more detailed knowledge is gained regarding the chemical and biological relevance of these materials.

However, several important criteria need to be met to formally establish extractives as suitable wood protectants. The most important one is the establishment of suitable test methods that can accurately assess the extractives; these might be adapted versions of the existing procedures, but the exploration of new methods to assess their efficacy should also be considered. Given the wide range of chemical compounds that encompass extractives, it is vital to tailor the methods that are suitable for the class of extractives in question. For example, light fraction terpenoids and other low-molecular-weight extractives are easily volatilized and readily isomerized by elevated heat [90], so methods will need to be developed that minimize the elevated temperatures, which are often commonplace in the workstream for standardized laboratory testing (oven drying, autoclaving, etc.). An obvious, but not to be overlooked, aspect of extractive utilization is solvent compatibility and the selection of a carrier that does not introduce negative environmental consequences. In addition to adequate evaluation methods, there should also be a concerted effort to develop analytical chemistry techniques to properly characterize, track, and quantify the extractives [37], so that their permanence in the intended substrate can be ascertained as a quality control measure to ensure proper amounts of bioactive extractives of interest are present in the host substrate and that those isoforms present are similar to those found naturally in the host species.

The insecticidal, or at least repellent, efficacy of extractives seems to be well established in the literature, and many chemical extractives have been shown to exhibit excellent biocidal, repellent, or antioxidant properties against a wide array of insect pests [17,91,92], either singly or in synergy with additional extractives or biocidal additives [93]. However, fungal efficacy, particularly that of wood decay fungi, is often much less promising. This is likely owing to the fact that decay fungi, particularly those who commonly encounter the host tree, are not unfamiliar with wood extractives and are not as easily deterred as insects. To alleviate this issue, the addition of co-biocides will likely yield a much more suitable and valuable product [94]. However, this slightly alters the regulatory pathways for product approval. A logical step towards this solution could be to evaluate additives currently listed in CFR 152.25(f)(2) as potential synergists with new and existing extractive evaluations.

More concrete and substantiated correlations between the increased resistance to termites and decay should be sought. Laboratory assays are commonly seen as the endpoint for exploratory testing, but long-term field testing is critical to assess how the entirety of fauna in the environment will respond to extractives in both in- and above-ground exposure. Using the studies summarized in Table 2, only 5% of these studies involved a field component. The typical monocultured soil bottle or plate-based assays are intended to be a rapid screening tool for ascertaining the efficacy of wood treatments at preventing mass loss due to the decay caused by an active isolate of decay fungi. However, so many other interacting variables can influence field-based tests and should be fully weighed in determining the suitability of extractives as wood protectants. Abiotic factors, such as ultraviolet (UV), thermal, and oxidative decomposition, need to be addressed and understood, as those data are required of modern wood protectants and should be expected to perform similarly. The majority of the current data presented in this subject area show promise, but much more work is needed in this area.

Extractive utilization as wood protectants has been the focus of many studies over the recent decades, but the defined mechanisms of activity are not well understood and should be the focus of future studies. The increasing body of work does indicate that plant extracts can and do illicit behavioral responses and cause the mortality of subterranean termites, but the adequate control of wood decay fungi is often lacking and needs further study. The removal and reuse of extractives removes the added difficulty of properly characterizing the extractive content throughout the woody tissue, which has been demonstrated to vary considerably. Additionally, the external effects of abiotic factors, such as UV, thermal, and oxidative decomposition, need to be fully considered in relation to their efficacy and use. The regulatory path for successful extractive-based biocides may include some combination of the more traditional regulatory paths used for wood preservatives and be augmented with some elements of the biopesticide acceptance criteria. The eventual goal is to provide effective, targeted, and long-lasting wood protectants, while providing a utilization path for woody biomass through extractive harvesting.

## Figures and Tables

**Table 1 insects-15-00069-t001:** Examples of some commercially available naturally durable wood species.

Common Name	Scientific Name	Type of Wood	Region	Durability Class		Reference
				Decay	Termite	
Eastern red cedar	*Juniperus virginiana*	Softwood	Temperate America	1–2	-	[11]
Alaska cedar	*Chamaecyparis nootkatensis*	Softwood	Temperate America	2	Susceptible	[11,12]
Western redcedar	*Thuja plicata*	Softwood	Temperate America	2	Susceptible	[11,12]
White cypress	*Callitris glaucophylla*	Softwood	Australia	2	Resistant	[13]
Redwood	*Sequoia sempervirens*	Softwood	Australia, Temperate America	2	Resistant	[11,13]
Teak	*Tectona grandis*	Hardwood	Tropical America, Asia	1	Moderately durable	[11,12]
Kokrodua	*Afrormosia elata*	Hardwood	Africa	1	Durable	[11]
Messmate	*Eucalyptus cloeziana*	Hardwood	Australia	1	Resistant	[13]
Bull oak	*Allocasuarina luehmannii*	Hardwood	Australia	1	Resistant	[13]
Black locust	*Robinia pseudoacacia*	Hardwood	Temperate America	1–2	Durable	[11,12]
Rosewood	*Dalbergia sissoo*	Hardwood	Asia	1	-	[11]
Red ironbark	*Eucalyptus sideroxylon*	Hardwood	Australia	1	Resistant	[11,13]
Chengal	*Neobalanocarpus heimii*	Hardwood	Asia	1	-	[11]
Jarrah	*Eucalyptus marginata*	Hardwood	Australia	2	Resistant	[13]
African blackwood (ebony)	*Dalbergia melanoxylon*	Hardwood	Africa	1	-	[11]
African padauk	*Pterocarpus soyauxii*	Hardwood	Africa	1	Durable	[11,12]
Sal	*Shorea robusta*	Hardwood	Asia	1–2	-	[11]
Idigbo	*Terminalia ivorensis*	Hardwood	Africa	2	Susceptible	[11,12]

## Data Availability

No new data were created or analyzed in this study. Data sharing is not applicable to this article.

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
