# Peer review of "Critical Review on the Use of Extractives of Naturally Durable Woods as Natural Wood Protectants"

_insects, 2024, doi:10.3390/insects15010069_

Round 1
Reviewer 1 Report
Comments and Suggestions for Authors
The authors review the background of and studies in which natural products are extracted from insect/fungal-resistant heartwoods and impregnated into non-resistant wood (presumably sapwood) to impart pest resistance to the latter.
Sources of these resistant heartwoods, however, is not deeply touched upon by the authors. It is not listed whether the “commercially available” wood species listed in Table 1 are indeed available as heartwood. Heartwood is found in old growth trees that are usually centuries old. Most of the commercially available heartwoods are leftovers from timbers downed from logging during the first half of the of the last century and are exceedingly rare and expensive, if available at all.
For the first studies (Rodrigues et al. 2012a, 2012b) listed in Table 2, mixtures of recovered sapwood and heartwood from Guyanese logging waste. Indeed the authors demonstrated antifungal and antitermital (rubrynolide from Sexonian rubra) when extractives were impregnated in susceptible wood. These woods are very expensive and are for indoor use (cabinetry, flooring, and furniture) https://www.woodworkerssource.com/lumber/latin-american-woods.html. As noted by the authors, many resistant woods are protected by CITES while others are endangered. It is therefore economically impractical to impregnate extractive from these woods or wood wastes into sustainable wood for outdoor use.
The authors should stress that these extractives be chemically characterized for active components that can be synthesized commercially to replacing currently used preservatives. One example might be rubrynolide from Sextonia (Rodriques et al. 2012b).

Author Response
Reviewer 1: I have addressed all of the substantial changes marked on the manuscript listed below and are presented as track changes in the revised manuscript.
Line 48: worldwide changed to lowercase.
Line 52: livestock fencing added to list of examples.
Line 53-54: complete sentence was removed and the authors agree.
Line 82: classification scale was removed and a more concise explanation of the durability rating scale was added.
Line 86-88: clarification was added to statement indicating that the IBC does not differentiate between old growth and second growth and that old growth timbers have been shown to contain greater proportions of heartwood extractives.
Line 111: “and age of the tree” added as recommended.
Line 133: Examples added to better define forest residues.
Many thanks for the editorial improvements.
Reviewer 2 Report
Comments and Suggestions for Authors
Title:
ok.
Abstract
Ok
Line 44:
Consider breaking down the information in this sentence for better readability.
Line 65:
Explicitly state the main thesis or objective of the review at this point.
Line 103:
Elaborate on the variability in extractive content and its role as the limiting factor.
Line 135:
Clarify and break down the description of extraction procedures into concise steps for better clarity.
Line 140:
Briefly explain the historical significance or outcomes of using pine tars in the 17th century for preserving wood on maritime vessels.
Line 144:
Specify examples or references for the solvents commonly used in modern extraction procedures.
Line 151:
Provide a brief explanation of vacuum-pressure impregnation for readers unfamiliar with the term.
Line 155:
Clarify the criteria used in laboratory and field tests to assess the efficacy of commercial wood preservatives and extractives as wood protectants.
Line 158:
Ensure consistency in terminology; use either "compounds" or "extractives" consistently throughout the text.
Line 160:
Briefly explain any abbreviations or classifications used in Table 2 to aid reader understanding.
Line 170-188:
Highlight the significance of the observed reduction in efficacy when stilbenes from pine, Osage orange, and eucalypt wood were impregnated into non-durable wood.
Emphasize the importance of understanding differences in binding between naturally and artificially impregnated wood.
Line 190-238:
Clarify the implications of regulatory hurdles, especially the cancellation of pentachlorophenol in 2021.
Consider providing specific examples or cases to illustrate the regulatory challenges.
Line 240-302:
· Reiterate the potential of extractive utilization as value-added forest products.
· Emphasize the need for tailored test methods and analytical chemistry techniques to assess and characterize extractives effectively.
· Suggest exploring potential synergists for improved efficacy against wood decay fungi.
· Encourage more long-term field testing to assess real-world effectiveness.
· Stress the importance of understanding abiotic factors like UV, thermal, and oxidative decomposition in evaluating extractives' performance.
· Acknowledge the gaps in understanding the mechanisms of activity and call for further research in this area.
General comments
After carefully reviewing the manuscript, I must commend the author for their skillful writing and overall presentation. However, I have identified several areas where the manuscript could be improved. These suggestions will help the author further enhance the manuscript's readability, structure, and impact.
Comments on the Quality of English LanguageMinor editing of the English language required.
Author Response
Reviewer 2:
Line 44:
Consider breaking down the information in this sentence for better readability. Done
Line 65:
Explicitly state the main thesis or objective of the review at this point. Done
Line 103:
Elaborate on the variability in extractive content and its role as the limiting factor. Some additional examples of factors that contribute to extractive variability were identified in lines 110-111.
Line 135:
Clarify and break down the description of extraction procedures into concise steps for better clarity. General description of solvent extraction procedure was added in line 153-155.
Line 140:
Briefly explain the historical significance or outcomes of using pine tars in the 17th century for preserving wood on maritime vessels. Added a paragraph to this effect.
Line 144:
Specify examples or references for the solvents commonly used in modern extraction procedures. This was done in conjunction with the edits to line 135.
Line 151:
Provide a brief explanation of vacuum-pressure impregnation for readers unfamiliar with the term. Two sentences were added in line 177-178 to provide background on the wood preservation process.
Line 155:
Clarify the criteria used in laboratory and field tests to assess the efficacy of commercial wood preservatives and extractives as wood protectants. Language added to line 173-175.
Line 158:
Ensure consistency in terminology; use either "compounds" or "extractives" consistently throughout the text. Compounds was replaced in the body of the text where referring specifically to extractives for clarity.
Line 160:
Briefly explain any abbreviations or classifications used in Table 2 to aid reader understanding. Language added in line 188-189.
Line 170-188:
Highlight the significance of the observed reduction in efficacy when stilbenes from pine, Osage orange, and eucalypt wood were impregnated into non-durable wood. Language added in lines 202-208 to address the two following suggestions.
Emphasize the importance of understanding differences in binding between naturally and artificially impregnated wood. See comment above.
Line 190-238:
Clarify the implications of regulatory hurdles, especially the cancellation of pentachlorophenol in 2021. Changed wording and added content to clarify
Consider providing specific examples or cases to illustrate the regulatory challenges.
Line 240-302:
- Reiterate the potential of extractive utilization as value-added forest products. See line 286
- Emphasize the need for tailored test methods and analytical chemistry techniques to assess and characterize extractives effectively. See line 291.
- Suggest exploring potential synergists for improved efficacy against wood decay fungi. Sentence added to line 308-309.
- Encourage more long-term field testing to assess real-world effectiveness. See line 318-320.
- Stress the importance of understanding abiotic factors like UV, thermal, and oxidative decomposition in evaluating extractives' performance.
- Acknowledge the gaps in understanding the mechanisms of activity and call for further research in this area. Language added to line 330.
General comments
After carefully reviewing the manuscript, I must commend the author for their skillful writing and overall presentation. However, I have identified several areas where the manuscript could be improved. These suggestions will help the author further enhance the manuscript's readability, structure, and impact.
Many thanks again for the thoughtful edits. I have addressed these in full and hope that the editors and reviewers are pleased with the resulting manuscript. Please let me know if you have any additional concerns.